# Incidence and Associations of Acute Kidney Injury after General Thoracic Surgery: A System Review and Meta-Analysis

**DOI:** 10.3390/jcm12010037

**Published:** 2022-12-21

**Authors:** Yang Yu, Shanshan Xu, Bing Yan, Xiaodong Tang, Honggang Zhang, Caifei Pan, Shengmei Zhu

**Affiliations:** 1Department of Anesthesiology, The First Affiliated Hospital, College of Medicine, Zhejiang University, Hangzhou 310000, China; 2Department of Anesthesiology, Haining People’s Hospital, Haining 314499, China

**Keywords:** acute kidney injury, thoracic surgery, incidence, postoperative, systematic reviews, meta-analysis

## Abstract

(1) Background: Acute kidney injury (AKI) is related to adverse outcomes in critical illness and cardiovascular surgery. In this study, a systematic literature review and meta-analysis was carried out to evaluate the incidence and associations of AKI as a postoperative complication of thoracic (including lung resection and esophageal) surgical procedures. (2) Methods: Adopting a systematic strategy, the electronic reference databases (PubMed, EMBASE, and Cochrane Library) were searched for articles researching postoperative renal outcomes that were diagnosed using RIFLE, AKIN or KDIGO consensus criteria in the context of a thoracic operation. A random-effects model was applied to estimate the incidence of AKI and, where reported, the pooled relative risk of mortality and non-renal complications after AKI. The meta-analysis is registered in PROSPERO under the number CRD42021274166. (3) Results: In total, 20 studies with information gathered from 34,826 patients after thoracic surgery were covered. Comprehensively, the incidence of AKI was estimated to be 8.8% (95% CI: 6.7–10.8%). A significant difference was found in the mortality of patients with and without AKI (RR = 2.93, 95% CI: 1.79–4.79, *p* < 0.001). Additionally, in patients experiencing AKI, cardiovascular and respiratory complications were more common (*p* = 0.01 and *p* < 0.001, respectively). (4) Conclusions: AKI is a common complication associated with adverse outcomes following general thoracic surgery. An important issue in perioperative care, AKI should be considered as a highly significant prognostic indicator and an attractive target for potential therapeutic interventions, especially in high-risk populations.

## 1. Introduction

With the advancement of medical science, the number of people diagnosed with pulmonary nodules or esophageal disease is steadily increasing [1,2]. Surgical tumor resection remains a mainstay of treatment and appears to be curative in most cases of primary cancer [3,4]. The practice of perioperative fluid restriction for the purpose of decreasing perioperative lung injury may increase the risk of renal insufficiency in patients undergoing thoracic surgery because of relative hypovolemia [3,5].

Acute kidney injury (AKI) is increasingly recognized as a common and serious complication after thoracic surgery, with incidence levels ranging from 2.2% to 35.3% [6,7,8], and it is related to long-term adverse consequences, including the progression to chronic kidney disease (CKD), as well as higher rates of cardiovascular disease and mortality [9,10,11,12]—even if renal function is clearly recovered at the time of discharge [5,8,12]. Hence, this might be significative of how, as a measurable signal of a prognostic indicator, postoperative AKI can be targeted for treatment.

In recent years, several meta-analyses associated with AKI have been performed in the context of cardiovascular and abdominal surgery [13,14,15,16,17,18]. The incidence of AKI after thoracic surgery is second only to that for cardiovascular surgery and general surgery [17,19], but no systematic review has been conducted on this subject.

For this reason, a systematic review and meta-analysis was conducted in order to summarize the incidence of AKI in the context of general thoracic surgery and explore the relationship between AKI and other postoperative complications.

## 2. Materials and Methods

The meta-analysis was conducted in accordance with PRISMA and MOOSE guidelines [20,21], and the study followed the guidance of the STROBE evaluation method [22]. The protocol for this study is registered in PROSPERO under the number CRD42021274166.

### 2.1. Study Selection

Studies were included if they met the following eligibility criteria: (1) they were retrospective or prospective observational studies that provided data on the incidence of AKI after general thoracic surgery; (2) AKI was defined by any of the following consensus definitions: RIFLE (risk, injury failure, loss, end stage), AKIN (acute kidney injury network), or KDIGO (kidney disease improving global outcomes) [23,24,25]. Exclusion criteria included studies in which the diagnosis of AKI was based on diagnostic codes or the need for renal replacement therapy and studies in which cardiovascular surgery was reported.

### 2.2. Search Strategy

We electronically searched PubMed, EMBASE, and the Cochrane Library for original articles published from database inceptions in peer-reviewed journals excluding case studies, letters, reviews, minutes, and summary publications. The most recent search was performed on 11 June 2022. We used the following terms mapped to standard medical subject headings and free-text words for literature searches: AKI, acute kidney injury, renal failure, kidney failure, lung surgery, thoracic surgery, and esophageal surgery. Moreover, we manually reviewed the reference lists of all included studies and those from relevant reviews and meta-analyses in order to identify additional studies. The search was not limited by any restrictions on language. Appendix A presents the detailed search strategy adapted for each database.

Two investigators (Y.Y. and S.X.) independently and sequentially reviewed the retrieved articles to determine their eligibility. Any disagreement between the review authors was resolved through consensus, or where necessary, by a third party (B.Y.) until a consensus was reached. EndNote 20.0 literature management software was used for the screening process.

### 2.3. Data Abstraction

The following information was obtained from the study using a structured data collection form: the title of the article, date of publication, country where the research was conducted, study design, diagnostic criteria for AKI, risk factors for AKI, incidence of AKI, incidence of severe AKI requiring renal replacement therapy (RRT), frequency of other postoperative complications (pulmonary complications including atelectasias, pneumonia, and respiratory failure, as well as cardiac complications including myocardial infarct, unstable arrhythmias, and congestive heart failure), length of hospital stay, and mortality (in-hospital mortality and 30-day mortality).

### 2.4. Study Quality

The quality of each study was independently assessed by two authors (B.Y. and H.Z.) using the Newcastle–Ottawa scale, which is one of the methods for measuring the quality of observational studies [26]. Any disagreement between the review authors was resolved through consensus, or where necessary, by a third party (X.T.). Studies were allocated a maximum of 9 points in value and defined as good (7–9), fair (4–6), or poor (0–3).

### 2.5. Statistical Analysis

Analyses were performed utilizing STATA version 16.0 (STATA Corp., College Station, TX, USA). Summary effect sizes were calculated as relative risks (RR) with 95% confidence intervals (95% CI) for dichotomous outcomes. To explain the presence of the possibility of between-study heterogeneity, data synthesis was conducted using a random-effects model, including the pooled effects of AKI in all studies as well as the relative risks of non-renal complications in patients with AKI. The *I*^2^ index (>50% indicating medium-to-high heterogeneity) and Cochran’s Q test (*p* < 0.05 for statistical significance) were utilized to determine the between-study heterogeneity. We conducted sensitivity analyses as well as a funnel plot analysis and the Egger test for publication bias.

## 3. Results

In total, our search identified 1260 records of which 113 records were removed as duplicates and a further 1091 records were excluded for not meeting our eligibility criteria upon the screening of the title and abstract. After a full-text examination, 18 studies met our selection criteria [5,6,7,8,9,10,11,12,27,28,29,30,31,32,33,34,35,36]. The reference lists for the bibliographies of the text articles were screened, which revealed two reports that met our inclusion criteria [19,37]. Finally, 20 studies including 34,826 patients in all reporting AKI outcomes in the setting of thoracic surgery were included in our analysis. The study identification and selection procedures are presented in Figure 1. Table 1 presents the clinical characteristics of all the included studies.

Of the 20 studies analyzed, 15 were retrospective observational studies [5,6,7,9,19,27,28,29,30,31,33,34,35,36,37], 4 were prospective observational studies [8,11,12,32], and 1 was a retrospective nested case-control study [10]. Two studies documented mixed thoracic surgical populations [19,29], thirteen documented lung surgical populations [5,7,9,11,12,27,30,32,33,34,35,36,37], and five were of esophageal surgical populations [6,8,10,28,31]. As for the quality of the included studies, 18 studies scored a 7 [5,6,7,8,9,11,12,19,27,28,29,30,31,32,34,35,36,37] and 2 scored a 9 [10,33] (Appendix A).

### 3.1. Incidence of AKI among Patients after Thoracic Surgery

From the meta-analysis, the pooled estimated incidence of AKI amongst the included studies was 8.8% (95% CI: 6.7–10.8%, *I*^2^ = 98.3%). From the 13 studies [5,6,8,9,11,12,19,27,28,29,34,35,36] where the stage of AKI was reported, 78.4% of patients with AKI had Stage 1 or RIFLE-R, 14.2% had Stage 2 or RIFLE-I, and 7.3% had Stage 3 or RIFLE-F (Appendix A). In contrast, the rates of postoperative RRT were low, ranging from 0 to 1.85% [6,8,9,11,12,19,28,29,30,31,32,35,36] (Table 1). Five studies reported renal outcomes in patients with AKI when stratified by the KDIGO AKI staging criteria [5] or AKIN grade [8,12,30,31]. The incidence of AKI decreased over time, and more than 70% of patients’ symptoms had resolved at discharge, with the recovery of their serum creatinine levels to normal values.

A subgroup analysis was carried out to explore the effects of different diagnostic criteria and surgical sub-categories on the incidence of AKI. There were no significantly different results found between the first two subgroups listed (*p* = 0.136 and *p* = 0.203, respectively, Table 2), while huge heterogeneity existed between the studies overall. Risk factors potentially related to the development of AKI were examined in some studies, and patients with pre-existing renal disease were significantly more likely to develop AKI. Twelve studies excluded patients with hemodialysis preoperatively, among which the preoperative eGFR < 15 mL/min/1.73 m^2^ [33,35,36]; only one study emphasized that the study population was derived from a cohort consisting of only patients with an eGFR ≥ 60 mL/min/1.73 m^2^ [19]. Three studies enrolled patients with normal renal function [7] (normal preoperative SCr and BUN [28], or excluding SCr > 2 mg/dL [12]). However, we did not find a difference in the pooled incidence rate at 9.3% (4.8–13.8%) versus at 9.4% (6.8–12.0%) versus at 4.6% (0.9–8.2%) (*p* = 0.092, Table 2) among the five studies examining unselected patients [8,30,32,34,37] or among the studies excluding hemodialysis preoperatively [5,6,9,10,11,19,27,29,31,33,35,36] and normal patients [7,12,28].

A meta-regression analysis of all the included studies demonstrated that the year of the study had no effect on the incidence of AKI (*p* = 0.162, *I*^2^ = 98.15%, R2 = 3.06%) among patients after thoracic surgery.

### 3.2. Mortality and Complications Risk of AKI in Patients after Thoracic Surgery

The incidence of mortality for patients with and without AKI after undergoing thoracic surgical procedures was reported in 10 studies [5,6,8,9,11,27,28,29,31,32] (Appendix A), among which 8 studies reported the in-hospital mortality, 2 studies reported the 30-day mortality, and only 1 reported the long-term mortality. The pooled RR of in-hospital or 30-day mortality among patients with AKI was 4.50 (RR = 4.50, 95% CI: 2.10–9.66, *p* < 0.001). While heterogeneity was statistically significant (range of RR 1.25–30.0), only four studies [27,28,29,32] found a significantly increased mortality rate among the patients experiencing AKI (Appendix A). When excluding RIFLE criteria [5,6,8,9,11,28,29,31,32], there was still a significant difference in the mortality of patients (RR =2.93, 95% CI: 1.79–4.79, *p* < 0.001) (Figure 2).

There were five studies [8,11,27,29,32] that reported the rates of postoperative pulmonary complications and cardiac complications, among which we counted the one with the most cases. The pooled effect demonstrated the incidence of cardiovascular and respiratory complications was significantly higher in patients with AKI than in those without AKI (RR = 2.26, 95% CI: 1.22–4.16, *p* = 0.01, Appendix A; RR = 3.29, 95% CI: 2.48–4.35, *p* < 0.001, Appendix A, respectively). Similarly, nine studies [6,8,9,11,27,29,31,32,36] reported the length of postoperative stay, which demonstrated a significant increase in the amounts of patients with postoperative AKI. We did not pool the effect size due to statistical heterogeneity.

### 3.3. Evaluation for Publication Bias and Sensitivity Analyses

The validity of the use of funnel plots as a means for detecting publication bias is affected by small studies of low quality, which can cause funnel-plot asymmetry [38]. Small-study effects can be observed due to real differences and the publication bias is only one of the potential reasons for this [39]. There are other possible biases, such as the intensity of intervention and differences in the underlying risk that may also lead to funnel-plot asymmetry. Therefore, in addition to a funnel plot (Appendix A), we conducted Egger’s regression test for quantitative data and small-study effects. No significant publication bias was found in our meta-analysis, *p* = 0.209. We also performed a sensitivity analysis by sequentially removing each study, which showed a stable effect for each individual study (Appendix A).

## 4. Discussion

In this meta-analysis, AKI was a relatively common complication in patients undergoing thoracic surgery with a pooled incidence rate of 8.8%, which is comparatively lower than for other types of surgery such as cardiac surgery (approximately 20–30%) [18], thoracic and abdominal aortic surgery (approximately 10–30%) [40], and major abdominal surgery (approximately 15%) [17]. In addition, the majority of patients with postoperative AKI had a case which was mild in severity, and less than 0.5% of patients in most studies experienced severe AKI requiring RRT [9,11,12,19,29,30,31,35,36]. Where renal recovery after AKI was reported, the AKI had resolved at discharge in more than 70% of patients [5,8,12,30], and in the majority of patients, long-term renal function returned to normal [8]. However, we were unable to obtain a pooled analysis of the rates of renal recovery due to differing definitions and methods of assessment. Nevertheless, we did observe from the studies that quite a few patients who experienced AKI had ongoing renal dysfunction or even required renal replacement therapy [6,8,28,32].

Despite the low incidence, patients who experience AKI after thoracic surgery still experience prolonged hospital stays and increased mortality [6,8,9,11,27,29,31,32,36]. Even though the RIFLE classification has several important limitations, it may affect the early diagnosis and treatment of AKI and increase mortality [41,42]; we did find a significant difference in the mortality of patients with or without AKI when excluding the study adapting RIFLE criteria [5,6,8,9,11,28,29,31,32]. Regretfully, none of the studies reported a correlation between the duration of renal impairment and the risk of mortality post thoracic surgery. Only one showed that the mortality risk tended to become progressively greater with the advancing stages of AKI [27], and another one found a significant difference in the long-term mortality of patients [32]. Interestingly, Samuel et al. found that surgical procedures were related to a lower risk of death; we suspect that the results are attributable to the selection of a healthy patient with the highest benefit for selective surgery and the resolution of the self-limited conditions fixed to selective surgery [43]. Moreover, even slight AKI is still related to adverse events, such as the development of cardiovascular and pulmonary complications. Thus, perioperative prophylaxis and treatment remain critically important to the prevention of postoperative renal injury.

On the other hand, substantial heterogeneity was found in the rate of AKI, even though our analysis focused on studies using consensus AKI definitions. Contrary to our expectation, the incidence of AKI did not significantly differ in our subgroup analysis, and the heterogeneity amongst all the studies could not be partly explained by these factors. Moreover, we found no significant correlation between the year of the study and the incidence of AKI post thoracic surgery in meta-regression analysis. The incidence of perioperative AKI was basically unchanged despite significant advances in diagnosis, surgical techniques, and perioperative management. [44]. Finding promising therapeutic advances to diagnose and treat perioperative AKI is still a challenge.

The perioperative period is when special pathophysiological conditions challenge the diagnosis of AKI, such as muscle injury, volume overload or hypovolemia, and the release of aldosterone and vasopressin from stress [45,46], which may well have an impact on the measurements of creatinine levels and oliguria [35,36,47,48]. Doctors and researchers have striven to find a “troponin-equivalent” marker for the precise identification of patients with AKI at an early stage, such as cystatin C, kidney injury molecule-1, or neutrophil gelatinase-associated lipocalin [49,50,51]. What is more, the urinary biomarkers tissue inhibitor of metalloproteinases-2 (TIMP-2) and insulin growth factor-binding protein 7 (IGFBP7) have been approved to help in identifying patients at a high risk for AKI in some interventional studies in cardiac and visceral surgery settings [52,53,54], as well as for improving the prediction levels of RRT and 30-day mortality after cardiac surgery [55]. The advantage is that they are able to detect kidney stress prior to injury or loss of function, allowing for much earlier therapy as compared to management guided by serum creatinine and urine output [54].

The mechanism of connection between surgery and AKI is still complex. The neuroendocrine response to hypotension, inflammation, and surgical trauma could probably damage kidney perfusion [29,34,56,57]. The reduction in renal blood flow and the reduction of renal oxygen supply lead to renal tissue hypoxia, which causes a cascade and further increases systemic inflammation [58,59]. AKI should be considered as a multi-organ system problem leading to dysfunction in the pulmonary, cardiac, neurologic, immunologic, and gastrointestinal systems, especially in high-risk groups [58].

The detrimental risk factors for AKI of a long anesthesia/surgery time, a complex surgical procedure, peri-operative hypotension and hypertension, blood loss, and ventilator-induced lung injury are well-recognized. In our review, a total of 16 studies reported AKI-associated risk factors using a multivariate or adjusted model. The reported risk factors for AKI are shown in Table 3. Old age [5,8,12,19,28,36], ASA 3 or 4 [12,27,30,32], higher body mass (BMI) [6,8,19,30,31], reduced baseline estimated glomerular filtration rate(eGFR) [9,19,29,35], low serum albumin concentration [6,29,36], diabetes mellitus (DM) [19,28,29,36], hypertension [9,10,11,19,30,32,36], the use of angiotensin-converting enzyme inhibitors/angiotensin receptor blockers (ACEI/ARB) [6,9,19,29,36], undergoing a thoracotomy procedure [9,11,29,30], and prolonged surgery time [10,12,27,32] were examined and identified. We found that the preoperative serum creatinine level was a significant indicator for AKI in several studies excluding hemodialysis [10,11,31,36]. Quantitative analysis was not possible due to the lack of individual data.

There are conflicting data on the application of ACEIs/ARBs to AKI. ACEIs/ARBs have been commonly prohibited preoperatively to prevent intraoperative hypotension [49,50,51,60,61]. While some other researchers have argued that the beneficial pleiotropic effects of ACEIs/ARBs go far beyond blood pressure reduction, ACEIs/ARBs can improve renal recovery or reduce the fibrotic processes of renal function impairment after AKI, which is associated with a better prognosis for patients with AKI [62,63]. Quite a few studies have identified that preoperative hypoalbuminemia was independently associated with AKI [6,29,36]. Li et al. discussed how serum albumin may have some reno-protective effects in improving renal perfusion, binding endogenous toxins and nephrotoxic drugs, and scavenging reactive oxygen species [64]. Wiedermann et al. performed a meta-analysis that determined hypoalbuminemia was an independent predictor of AKI and AKI- related deaths [65]. As for the colloid infusion during surgery, there is no consensus [66], while hydroxyethyl starch solutions should be used with caution in high-risk patients undergoing thoracic surgery and have been discouraged in recent years [6,9,29].

Developing successful therapies to treat AKI has always been an elusive effort. Numerous agents (N-Acetylcysteine, the lipid-lowering 3-hydroxy-3-methylglutaryl coenzyme, dexmedetomidine, and so on) have shown promise [67,68,69], while clinical effectiveness varies between studies and the current evidence does not confirm the effectiveness of the agents in the treatment or prevention of AKI [30,70,71]. A single measure for prevention and therapy measures for AKI do not work well in clinical practice. A combination of treatments—including nutritional support [72] and glycemic control [25], minimizing nephrotoxic medication exposure [73], and hemodynamic optimization [29,34,74,75]—have largely been studied. Recently, new biomarkers of AKI have been discovered and validated [49,50,51,55]. In general, a combination of the biomarkers (urinary TIMP-2 and IGFBP7) for the early detection of perioperative kidney damage and accelerated intervention schemes seems to be the basis for AKI prophylaxis and treatment in surgical settings [52,55].

There are several limitations to our meta-analysis. Firstly, there are statistical heterogeneities in our meta-analysis. Subgroup analyses and meta-regression analysis cannot adequately explain the considerable heterogeneity. Secondly, AKI diagnosis in our analysis was mainly based on changes in serum creatinine and urine output, and there is limited data on novel biomarkers of AKI. Lastly, all the studies we included were observational studies. Thus, we merely demonstrated an association between AKI and increased complications after thoracic surgery, and the causality still needs to be confirmed by a large number of clinical trials or population-based studies of high quality.

In summary, AKI is a common complication following general thoracic surgery and is associated with an increased risk of further non-renal postoperative complications and mortality. Despite the progress in perioperative management, the incidence of AKI in patients does not appear to have improved, suggesting the need for a greater attention to AKI following thoracic surgery, especially in particular populations.

## Figures and Tables

**Figure 1 jcm-12-00037-f001:**
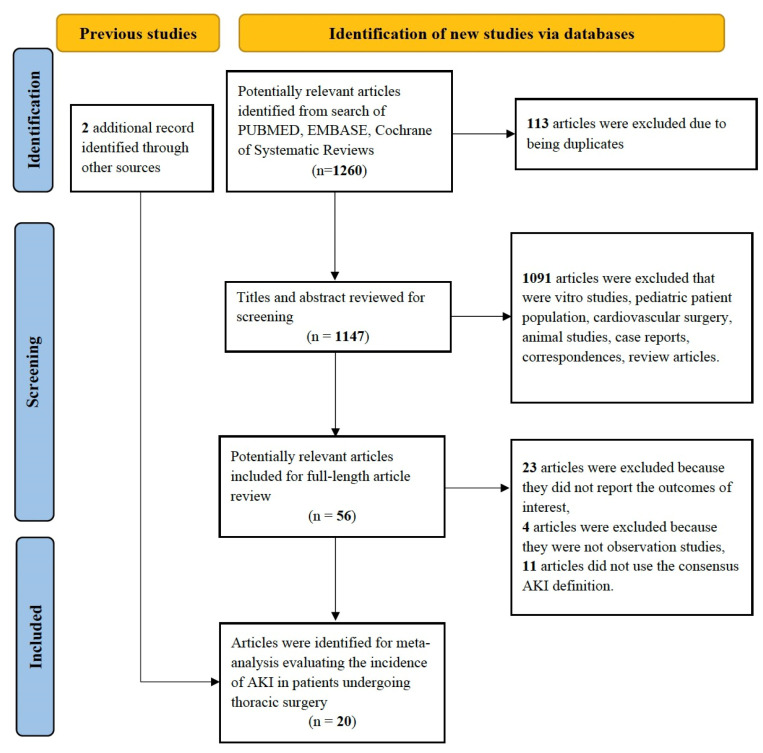
Study identification and selection procedure.

**Figure 2 jcm-12-00037-f002:**
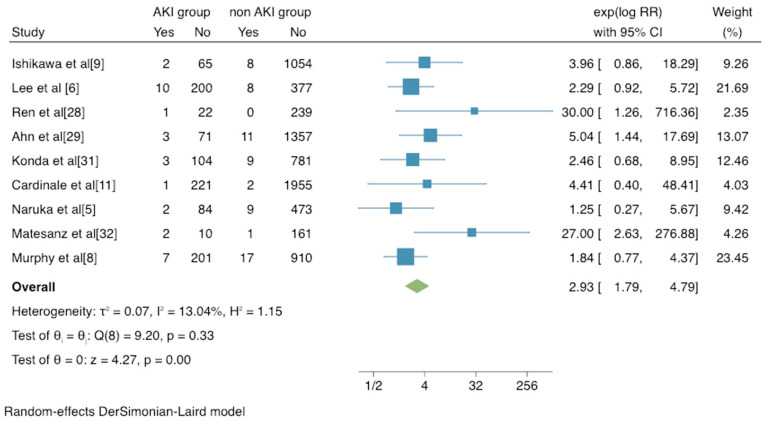
Forest plots of postoperative mortality in patients with/without postoperative AKI. Blue square represents RR of each study; green square represents the pooled RR of all studies. AKI, acute kidney injury; AKIN, acute kidney injury network; CI, confidence interval; RR, relative risks.

**Table 1 jcm-12-00037-t001:** Main characteristics of the studies included in the analysis.

Study (Year)	Country	Setting	Patients Type (Non-AKI vs. AKI)	Number of Patients	Number with AKI	AKI Incidence	RRT Incidence	AKI Definition
Licker et al. (2011) [27]	Switzerland	Lung	Excluding hemodialysis Age: 63 (11) vs. 66 (9) Preoperative eGFR: 83 (23) vs. 75 (21) DM: 9.3% vs. 23.3% CAD: 10.5% vs. 12.5%	1345	91	0.068	--	RIFLE
Ishikawa et al. (2012) [9]	Canada	Lung	Excluding hemodialysis Age: 61 (15) vs. 67 (10) Preoperative eGFR: 74 (22) vs. 62 (23) Preoperative SCr: 79.56 (17.68) vs. 97.24 (35.36) DM: 9% vs. 19% CAD: 11% vs. 19%	1129	67	0.059	1/1129 (0.09%)	AKIN
Lee et al. (2014) [6]	Korea	Esophageal	Excluding hemodialysis Age: 61.7 (8.2) vs. 62.6 (8.2) Preoperative eGFR < 60 mL/min/1.73 m^2^: 3.1% vs. 4.8% Preoperative SCr: 70.72 (61.88–79.56) vs. 70.72 (61.88–79.56) DM: 15.1% vs. 21.4% CAD: 1.3% vs. 0.5%	595	210	0.353	11/595 (1.85%)	AKIN
Ren et al. (2015) [28]	China	Esophageal	Normal Age: 63 (15) vs. 74 (12) Preoperative SCr: 78 (11) vs. 75 (13) DM: 18% vs. 39.1% CAD: 23% vs. 26.1%	362	23	0.064	2/362 (0.55%)	KDIGO
Assaad et al. (2015) [12]	USA	Lung	Excluding SCr > 2 mg/dL Mean age of all: 67 (from 54 to 83)	40	3	0.075	0	AKIN
Grams et al. (2016) [19]	USA	Thoracic	Excluding hemodialysis Mean age of all: 64 (10) Preoperative eGFR < 60 mL/min/1.73 m^2^: 1.2% DM: 27% CAD: 27%	11,779	1413	0.120	23/11779 (0.2%)	KDIGO
Ahn et al. (2016) [29]	Korea	Thoracic	Excluding hemodialysis Age: 59.4 (12.5) vs. 64.5 (10.3) Preoperative eGFR: 92 (21) vs. 93 (41) DM: 13% vs. 39%	1442	74	0.051	2/1142 (0.18%)	AKIN
Moon et al. (2016) [30]	USA	Lung	Unselected Mean age of all: 66 (from 59 to 73)	1207	98	0.081	0	AKIN
Konda et al. (2017) [31]	USA	Esophageal	Excluding hemodialysis Age: 60 (10) vs. 63 (9) Preoperative SCr: 79.6 (17.7) vs. 91.1 (26.5) DM: 13.3% vs. 26.2% CAD: 15.4% vs. 23.4%	897	107	0.119	0	AKIN
Wang et al. (2017) [10]	China	Esophageal	Excluding hemodialysis Age: 63 (8) vs. 63 (9) Preoperative SCr: 76 (13) vs. 85 (23) DM: 8.3% vs. 11.8% CAD: 4.9% vs. 7.8%	2094	51	0.024	--	KDIGO
Cardinale et al. (2018) [11]	Italy	Lung	Excluding hemodialysis Age: 62 (10) vs. 68 (9) Preoperative eGFR: 102 (84–117) vs. 83 (67–103) Preoperative SCr: 67.18 (56.58–78.67) vs. 81.33 (68.07–98.12) DM: 8% vs. 8% CAD: 4% vs. 20%	2179	222	0.102	5/2179 (0.23%)	AKIN
Naruka et al. (2019) [5]	UK	Lung	Excluding hemodialysis Old (>60 years):58.7% vs. 75.6	568	86	0.151	--	KDIGO
Matesanz et al. (2019) [32]	Spain	Lung	Unselected Age: 65 (56–70) vs. 73 (64–77)	174	12	0.069	2/174 (1.15%)	AKIN
Oh et al. (2019) [33]	Korea	Lung	Excluding hemodialysis Aged 19 yr or older	2872	140	0.049	--	KDIGO
Garutti et al. (2019) [37]	Spain	Lung	Unselected Aged 19 yr or older	174	12	0.069	--	AKIN
Murphy et al. (2020) [8]	UK	Esophageal	Unselected Mean age of all: 64.2 ± 9.2	1135	208	0.183	10/1135 (0.88%)	AKIN
Meng et al. (2020) [35]	China	Lung	Excluding hemodialysis Age: 59.8 (10.6) vs. 58.8 (10.7) Preoperative eGFR < 30 mL/min/1.73 m^2^: 0.3% vs. 9.7% DM: 17.4% vs. 32.3% CAD: 11.4% vs. 12.9%	1393	31	0.022	1/1393 (0.07%)	KDIGO
Kim et al. (2020) [34]	Korea	Lung	Unselected	1031	63	0.061	--	AKIN
Zhao et al. (2021) [36]	China	Lung	Excluding hemodialysis Age: 58 (51–65) vs. 63 (56–69) Preoperative SCr: 70 (60–82) vs. 75 (62–90) DM: 11% vs. 23% CAD: 5% vs. 13%	3862	205	0.053	0	KDIGO
Wu et al. (2021) [7]	China	Lung	Normal	548	12	0.022	--	AKIN

Abbreviations: AKI, Acute kidney injury; AKIN, acute kidney injury network; CAD, coronary artery disease; DM, Diabetes mellitus; eGFR, estimated glomerular filtration rate; KDIGO, kidney disease improving global outcomes; RIFLE, risk, injury, failure, loss of kidney function, and end-stage kidney disease; RRT, renal replacement therapy; SCr, serum creatinine.

**Table 2 jcm-12-00037-t002:** Subgroup analysis of the proportion of patients who developed postoperative AKI.

	Number of Studies	Number of Patients	AKI Incidence	95% CI	*I* ^2^	*p*
All patients	20	34,826	0.088	0.067–0.108	98.3%	
AKI definition						0.136
RIFLE	1	1345	0.068	0.054–0.081	--	
AKIN	12	10,551	0.103	0.071–0.135	97.4%	
KDIGO	7	22,930	0.068	0.035–0.101	99.1%	
Surgical type						0.203
Lung	13	16,522	0.066	0.050–0.081	94.2%	
Esophageal	5	5083	0.148	0.055–0.240	99.2%	
Thoracic	2	13,221	0.086	0.019–0.153	99.1%	
Preoperative renal function						0.092
Unselected	5	3721	0.093	0.048–0.138	95.3%	
Excluding hemodialysis	12	30,155	0.094	0.068–0.120	98.8%	
Normal	3	950	0.046	0.009–0.082	79.3%	

Abbreviations: AKI, Acute kidney injury; AKIN, acute kidney injury network; KDIGO, kidney disease improving global outcomes; RIFLE, risk, injury, failure, loss of kidney function, and end-stage kidney disease; CI, confidence intervals.

**Table 3 jcm-12-00037-t003:** Reported risk factors for AKI in patients undergoing thoracic surgery.

Study (Year)	Setting	Patients Type	Risk Factors for AKI
Licker et al. (2011) [27]	Lung	Excluding hemodialysis	ASA 3 or 4, low FEV1, use of vasopressors, prolonged anesthesia time
Ishikawa et al. (2012) [9]	Lung	Excluding hemodialysis	hypertension, peripheral vascular disease, low eGFR, use of ARB, intraoperative hydroxyethyl starch administration, thoracotomy procedure
Lee et al. (2014) [6]	Esophageal	Excluding hemodialysis	BMI, low serum albumin level, use of ACEI or ARB, intraoperative hydroxyethyl starch administration, postoperative 2-day CRP
Ren et al. (2015) [28]	Esophageal	Normal	elderly, DM, intraoperative hypotension
Assaad et al. (2015) [12]	Lung	Normal	elderly, ASA 3 or 4, prolonged surgery time
Grams et al. (2016) [19]	Thoracic	Excluding hemodialysis	elderly, male, African American, higher BMI, hypertension, DM, lung disease, malignancy, low eGFR, use of ACEI/ARB, diuretic use, later timing of surgery during the hospital stay
Ahn et al. (2016) [29]	Thoracic	Excluding hemodialysis	use of ACRI/ARB, open thoracotomy, pneumonectomy/esophagectomy, DM, cerebrovascular disease, low serum albumin level, decreased renal function(eGFR < 60 mL/min/1.73 m^2^)
Moon et al. (2016) [30]	Lung	Unselected	BMI, male, ASA 3 or 4, hypertension, smoking status, thoracotomy procedure
Konda et al. (2017) [31]	Esophageal	Excluding hemodialysis	higher BMI, a number of comorbidities, high preoperative creatinine level
Wang et al. (2017) [10]	Esophageal	Excluding hemodialysis	preoperative serum creatinine level, duration of surgery, smoking status, hypertension
Cardinale et al. (2018) [11]	Lung	Excluding hemodialysis	hypertension, preoperative serum creatinine level, forced vital capacity, preoperative NT-proBNP, pneumonectomy, intraoperative blood loss
Naruka et al. (2019) [5]	Lung	Excluding hemodialysis	60 years or older
Matesanz et al. (2019) [32]	Lung	Unselected	hypertension, ASA 3 or 4, prolonged surgery time, plasma IL-6 level at 6 h after surgery
Murphy et al. (2020) [8]	Esophageal	Unselected	elderly, male, increased BMI, dyslipidemia
Meng et al. (2020) [35]	Lung	Excluding hemodialysis	intraoperative urine output < 0.8 mL/(kg·h), preoperative Hb ≤ 120.0 g/L, preoperative eGFR < 30 mL/min/1.73 m^2^)
Zhao et al. (2021) [36]	Lung	Excluding hemodialysis	elderly, hypertension, DM, use of ACEI/ARB, preoperative serum albumin and creatinine level, blood loss, intraoperative lowest MAP

Abbreviations: ACEI/ARB, angiotensin-converting enzyme inhibitor/angiotensin II receptor blockers; AKI, acute kidney injury; ASA, American Society of Anesthesiologists; BMI, body mass index; CRP, C-reactive protein; DM, diabetes mellitus; FEV1, forced expiratory volume in 1 s; eGFR, estimated glomerular filtration rate; MAP, mean arterial pressure; NT-proBNP, N-terminal pro brain natriuretic peptide.

## Data Availability

The datasets used and/or analyzed during the current study are available from the corresponding author on reasonable request.

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
