# Peer review of "Incidence and Associations of Acute Kidney Injury after General Thoracic Surgery: A System Review and Meta-Analysis"

_jcm, 2022, doi:10.3390/jcm12010037_

Round 1

Reviewer 1 Report

Dear Dr. Yu, your paper „Incidence and associations of acute kidney injury after general thoracic surgery: a system review and meta-analysis” focuses on a clinical important aspect in daily clinical practice. The given data are presented in an appropriate manner and a good understandably writing style. Nevertheless, sometimes advanced data might improve impact of the paper:

1)      Regarding mortality: do the authors have any information if mortality differed between pts. with short AKI (0-7 days after thoracic surgery) and those with persisting AK, resp. AKD (7-90 days post thoracic surgery)

2)      what was the precise definition of “resolved AKI at discharge” (please compare discussion section, line 217)? please state on that

3)      regarding the statement “…troponin-equivalent marker”… (discussion section line 248-250): the FDA approved biomarker “TIMP*IGFBP7 (NephroCheck) has shown its impact even in some interventional studies in cardiac as well as visceral surgery settings (Zarbock A, et al Prevention of Cardiac Surgery-Associated Acute Kidney Injury by Implementing the KDIGO Guidelines in High-Risk Patients Identified by Biomarkers: The PrevAKI-Multicenter Randomized Controlled Trial. Anesth Analg. 2021 Aug 1;133(2):292-302. doi: 10.1213/ANE.0000000000005458. PMID: 33684086. ; Meersch M, et al. Prevention of cardiac surgery-associated AKI by implementing the KDIGO guidelines in high risk patients identified by biomarkers: the PrevAKI randomized controlled trial. Intensive Care Med. 2017 Nov;43(11):1551-1561. doi: 10.1007/s00134-016-4670-3. Epub 2017 Jan 21. Erratum in: Intensive Care Med. 2017 Mar 7;: PMID: 28110412; PMCID: PMC5633630.; Göcze I, et al. Biomarker-guided Intervention to Prevent Acute Kidney Injury After Major Surgery: The Prospective Randomized BigpAK Study. Ann Surg. 2018 Jun;267(6):1013-1020. doi: 10.1097/SLA.0000000000002485. PMID: 28857811.) Furthermore it showed its  relevance for prediction of post-surgery mortality (Esmeijer, K., et al. The predictive value of TIMP-2 and IGFBP7 for kidney failure and 30-day mortality after elective cardiac surgery. Sci Rep 11, 1071 (2021). https://doi.org/10.1038/s41598-020-80196-2) thus mentioning the impact/relevance of this as well some other biomarker might strengthen quality of this paper

4)      do the authors have any information on pre-operative eGFR, resp. what was the threshold eGFR (= pre-existing renal disease (please see page 7, line 156), which triggered post-operative AKI – an information which might have robust clinical impact;

5)      what was the definition of “normal” (compare table 2) ? eGFR > 60 ml/min? no albuminuria?

Author Response

Dear Reviewer,

Thank you for your letter and for the reviewers’ comments concerning our manuscript entitled “Incidence and Associations of Acute Kidney Injury after General Thoracic Surgery: A System Review and Meta-Analysis” (Submission ID: jcm-2043398). Those comments are all valuable and very helpful for revising and improving our paper, as well as the important guiding significance to our researches. We have studied comments carefully and have tried our best to revise our manuscript. Attached please find the revised version, which we would like to submit for your kind consideration. These changes will not influence the content and framework of the paper. Once again, thank you very much for your comments and suggestions.

Reviewer 2 Report

This is a systematic review on the incidence of AKI after thoracic surgery. The topic is of interest. However, there were many concerns regarding their data.

Major comments

1.      Regarding “3.3 Risk factors for AKI in patients undergoing thoracic surgery”, this is just a narrative review of literatures and this part does not qualify as a part of systematic review or meta-analysis.

2.      In the abstract, they wrote that RR of mortality among those with and without AKI was 2.93 (1.79-4.79). The 95% CI did not cross the boundary of 1 and the difference should be significant but the p values was 0.326. This does not make sense.

3.      Looking through all the data, there were significant heterogeneity among studies. They should present more detail about inclusion and exclusion criteria of the study population in Table 1 (For example, age, pre-operative eGFR, proportion of patients with CV diseases, DM etc).

4.      Page 8, line 177, log RR was 1.07 (0.58-1.57) and it did not cross the boundary of 0. The difference should be statistically significant, but the p values was 0.33. This does not make sense.

5.      What is the definition of short-term mortality? In hospital mortality or 30-day mortality?

6.      What is the definition of cardiovascular and respiratory complications?

7.      Regarding “3.4 Evaluation for publication bias”, the distribution of point of estimate for small-sized studies were apparently right-skewed. Was their analysis with p values of 0.209 correct?

Minor comments

1.      Page 3, line 118 Funnel ploy-> funnel plot

2.      Page8, line 185, not only p values, but log RR with 95% CI should be presented.

3.      Page 11, line 231, important prevent-> important to prevent

Author Response

(The authors gave the same response as above.)

Reviewer 3 Report

dear authors, this is an interesting work, however there are some issues that need clarification. 

In a random effects model, the pooled relative risk of hospital or 30-day mortality was higher for patients with AKI compared to patients without AKI

Is the risk of mortality the same with AKI population or is it higher due to the combined risk from surgery?

How do you explain that a significantly higher rate of cardiovascular and respiratory complications was noted in patients who developed AKI postoperatively compared to patients that did not.

You identified most factors that may be associated to AKI, but you do not discuss on any preventive measures or any post-operative supportive and treatment measures that those studies discussed. It is important to present any available evidence on the management of those patients.

Author Response

(The authors gave the same response as above.)

Round 2

Reviewer 2 Report

The manuscript has been improved. My only comment is that the definitions of short-term mortality and cardiopulmonary complications should be described in methods rather than in discussion. 
